# Age-Differentiated Leadership and Healthy Aging at Work: Evidence from the Early Stages of the COVID-19 Pandemic

**DOI:** 10.3390/ijerph182312509

**Published:** 2021-11-27

**Authors:** Ryszard J. Koziel, Jack C. Friedrich, Cort W. Rudolph, Hannes Zacher

**Affiliations:** 1Department of Psychology, Saint Louis University, St. Louis, MO 63103, USA; ryszard.koziel@slu.edu (R.J.K.); jack.friedrich@slu.edu (J.C.F.); 2Wilhelm Wundt Institute of Psychology, Leipzig University, 04109 Leipzig, Germany; hannes.zacher@uni-leipzig.de

**Keywords:** healthy aging, work ability, age-differentiated leadership, COVID-19

## Abstract

Little is known about the relative influence of age-differentiated leadership on healthy aging at work. Likewise, the age-conditional influence of age-differentiated leadership is understudied, and especially so in the context of the COVID-19 pandemic. Using a three-wave longitudinal study, we examined the role that age-differentiated leadership plays in the prediction of work ability, as measured three times over six months (*n* = 1130) during the early stages of the COVID-19 pandemic in Germany (i.e., December 2019, March 2020, and June 2020). The results suggest that although there were no systematic changes in work ability on average, there was notable within-person variability in work ability over time. Additionally, we find that a balanced approach to age-differentiated leadership that considers the needs of both older and younger employees matters most and complements the positive influence of leader–member exchange for predicting within-person variability in work ability. We also find that older employees’ work ability benefits from an approach to age-differentiated leadership that considers older employee’s needs, whereas younger employees’ work ability especially benefits from leader–member exchange and a balanced approach to age-differentiated leadership. Overall, these results provide initial support for the idea that an age-differentiated approach to leadership is important when considering healthy aging at work.

## 1. Introduction

The implications of the aging of the global workforce are well-documented [1]. Along with an on-average older workforce, organizations are experiencing an increase in the age diversity of their workers [2]. This increased age diversity comes with both opportunities (e.g., encouraging the potential for knowledge sharing at work [3]) and challenges (e.g., managing employee health to ensure long term employability [4]). At the same time, the COVID-19 pandemic has dramatically altered the way in which work is conducted, with some suggesting that higher employee age may be a risk factor for diminished wellbeing at work in the face of the pandemic [5,6]. Despite this, research on the experience of healthy aging at work during the pandemic remains scarce thus far [7].

One approach that has been proposed to address the challenges of an increasingly age-diverse workforce and to support healthy aging at work is the adoption of age-differentiated approaches to the design of work systems [8,9], including age-differentiated leadership [10,11]. To this end, researchers have defined three subdimensions of age-differentiated leadership [12]: leadership focused on younger employees (e.g., giving younger employees the support that they need), leadership focused on older employees (e.g., involving older employees in the discussion of upcoming changes at work), and an approach to leadership that balances the needs of both younger and older employees (e.g., promoting a positive “togetherness” between younger and older employees). Importantly, owing to the definition of these subdimensions, leaders can theoretically display all three forms of age-differentiated leadership simultaneously [12].

The promise of age-differentiated work systems and approaches to leadership is that by tailoring specific aspects of the work environment to individual needs and motives that may vary across the lifespan, people are better able to maintain person–environment fit over time [13]. As a result, people who experience age-differentiated leadership throughout their career may be better able to sustain successful and healthy aging at work. To this end, it has long been known that there are potential salutogenic effects associated with leadership [14,15]. We focus here on work ability as an assumed outcome of age-differentiated leadership because it constitutes a relevant indicator of the notion of healthy aging at work [16]. Work ability comprises employees’ perceived ability to meet the physical, mental, and interpersonal demands of their jobs [17,18].

### 1.1. Healthy Aging at Work

Healthy aging can be defined as “...the process of developing and maintaining the functional ability that enables well-being in older age” [19] (p. 28). Extending this idea, healthy aging *at work* has recently been defined as “…a motivational life-span process during which workers develop, maintain, or regain functional ability, comprised of the interplay, or fit between individual and environmental factors, which enables high well-being and resilience when workers are older” [16] (p. 76). In terms of the consideration of healthy aging at work in the present study, we have conceptualized environmental factors (i.e., leadership) that are associated with higher wellbeing (i.e., work ability) for people of different ages. Leaders serve an important role in supporting their employees and have a primary responsibility for enacting their organization’s “duty of care” (i.e., the responsibility to maintain the health and wellbeing of their employees [15]). Thus, in many ways, leaders are the conduit through which organizations can support healthy aging at work. Especially in times of crisis, such as in the face of a global pandemic, understanding the role that leadership plays in bolstering the health and wellbeing of employees, and particularly processes that may encourage healthy aging at work, is of paramount importance [20,21].

### 1.2. Age-Differentiated Leadership and Employee Work Ability

As suggested, the positive and negative influence that leaders can exert on their employees’ health and wellbeing has long been understood [14,15]. More recently, the concept of age-differentiated leadership has emerged as a novel way of considering how to best lead members of the increasingly age-diverse workforce. The model of age-differentiated leadership suggests that leadership behaviors directed at the various changing needs and motives of younger and older followers are important for the management of age-diverse workgroups [11,12]. For example, research based on the lifespan development perspective [22] suggests that younger workers are more strongly motivated by extrinsic growth goals (e.g., advancement, promotion), whereas older workers are more focused on intrinsic goals (e.g., autonomy, use of skills [23]). Theorizing on the beneficial effects of age-differentiated leadership assumes that leaders who, through their actions, meet the age-specific needs and motives of their followers are more likely to promote healthy aging. Thus far, however, little research has focused on the relationship between age-differentiated leadership and employee health and wellbeing. One exception to this is research which found that leadership behaviors that encourage open and positive attitudes toward age and ageing at work as well as an openness to participation and cooperation regardless of employee age were positively associated with employee work ability [24]. Building on these findings, research has demonstrated that age-differentiated leadership has a positive influence on employee health, reduces turnover intention, and increases performance, and that such relationships are observed across occupations and industries (e.g., nursing, call centers, production) [12]. Similar conclusions regarding the role of age-differentiated leadership have been reached by other scholars [25], who offer that “…leadership that is sensitive towards the specific needs of older followers seems to be critical for maintaining high work motivation and performance with increasing age of followers” (p. 4).

Thus, although there is some preliminary evidence for the utility of age-differentiated leadership for supporting employee wellbeing, we still do not have a clear picture of how it operates, and especially so in the broader context of other leadership phenomena. To this end, leadership researchers have suggested that it is important to demonstrate the influence of novel leadership constructs above and beyond established leadership constructs [26]. Thus, we first aim to consider the incremental predictive ability of age-differentiated leadership above and beyond LMX, which refers to the quality of the dyadic exchange relationship developed between a leader and a follower [27]. Meta-analytic research [28] has shown that LMX represents a key mediator of associations between four major leadership behaviors (i.e., consideration, initiating structure, contingent reward, transformational leadership) and follower performance (i.e., task performance, organizational citizenship behavior). 

Additionally, given its focus on balancing the needs of employees across the lifespan, all employees, regardless of their age, should benefit from leadership that balances a focus on both older and younger employees and on their respective needs. Moreover, according to recent conceptualizations of age-differentiated leadership, people of different ages should benefit when leadership is matched to the needs of their age. Specifically, younger employees should be more likely to benefit from age-differentiated leadership that focuses on younger employees, whereas older employees should be more likely to benefit from age-differentiated leadership that focuses on older employees [24,25]. 

### 1.3. Summary of Study Objectives

The present study was conducted to contribute to the literature on leadership and healthy aging at work in three important ways. First, as suggested, we know little about how healthy aging processes and leadership have co-occurred with one-another during the COVID-19 pandemic. The present study offers an important in situ look at these relationships during this especially challenging time for leaders and employees. Second, despite its promise, little is known about the relative effects of age-differentiated leadership on healthy aging at work above and beyond established leadership constructs (i.e., leader–member exchange, LMX [28]). Finally, although theoretically different forms of age-differentiated leadership should benefit older and younger employees, respectively, the age-conditional influence of age-differentiated leadership is understudied. Thus, we conducted this study to address three related goals. First, and most broadly, to understand how healthy aging at work and leadership unfold during the early stages of the COVID-19 pandemic. Second, to establish the relative importance of age-differentiated leadership against LMX for explaining variability in employees’ work ability. Third and finally, to consider the age-conditional influence of age-differentiated leadership for explaining variability in employees’ work ability over time. 

### 1.4. Hypotheses

Considering these objectives and drawing support from the theorizing reviewed above, the following hypotheses were tested in the present study:

**Hypothesis** **1.**
*Age-differentiated leadership predicts work ability incremental to leader–member exchange.*


**Hypothesis** **2.**
*A balanced approach to age-differentiated leadership benefits the work ability of employees of all ages.*


**Hypothesis** **3.**
*Older (younger) workers benefit more than younger (older) workers from age-differentiated leadership focused on older (younger) workers.*


## 2. Materials and Methods

### 2.1. Transparency and Openness

In service of transparency and openness, the de-identified data on which the study conclusions are based, the analytic code needed to reproduce the analyses in R, and complete results of all focal as well as supplemental analyses are available in our online appendix: https://osf.io/2bgwn/.

The data reported in this manuscript were collected as part of a larger longitudinal data collection. Thus far, four articles have been published based on the same dataset, with completely different research questions and completely different sets of substantive variables [29,30,31,32]. 

### 2.2. Ethics

This study was approved by the ethics board of Leipzig University (Protocol ID#: 2019.06.27_eb_17).

### 2.3. Participants

Data were obtained in a longitudinal study with three waves of data collection and time lags of three months (Time [T] 1 = beginning of December 2019, T2 = beginning of March 2020, T3 = beginning of June 2020). At T1, *n =* 2976 started the survey; the sample considered here consists of *n* = 1145 full-time employed workers in Germany who provided at least partial responses to demographic and/or substantive variables T1 to T3. Table 1 considers demographic characteristics of this sample compared to T1-only responders. Of note, owing to observed missing data on the substantive variables of interest, the focal models presented herein are based on *n =* 1133. 

### 2.4. Measures

Complete scales including all items used to measure our focal constructs can be found in our online appendix: https://osf.io/2bgwn/.

#### 2.4.1. Age-Differentiated Leadership

We used a sixteen-item scale to measure age-differentiated leadership at T1 to T3 [12]. Responses were collected on a 7-point response scale (ranging from *very strongly disagree* to *very strongly agree*). Participants were asked, “In the last three months, my supervisor…” and then to respond to items that reflect three forms of age-differentiated leadership, including leadership focused on younger employees (example item: “…offered younger employees opportunities to advance their professional development”), leadership focused on older employees (example item: “…dealt with the strengths and weaknesses of older employees when planning their work”), and a balanced approach to leadership that considers both younger and older employees (example item: “…offered older employees the same opportunities to acquire new knowledge and skills as younger ones”). Across all three dimensions, this scale demonstrated acceptable levels of reliability at each time point (younger: α_range_ = 0.944–0.951, ω_range_ = 0.943–0.951, AVE_range_ = 0.807–0.829; older: α_range_ = 0.942–0.948, ω_range_ = 0.942–0.947, AVE_range_ = 0.765–0.783; balanced: α_range_ = 0.957–0.963, ω_range_ = 0.957–0.962, AVE_range_ = 0.763–0.785).

#### 2.4.2. Leader–Member Exchange

LMX was measured at T1 to T3 with a seven-item short scale [27]; see also [33]. Example items are “How well does your leader understand your job problems and needs?” (5-point response scale ranging from *not a bit* to *a great deal*) and “Regardless of how much formal authority he/she has built into his/her position, what are the chances that your leader would use his/her power to help you solve problems in your work?” (5-point response scale ranging from *none* to *very high*). This scale demonstrated acceptable levels of reliability at each time point (α_range_ = 0.933–0.938, ω_range_ = 0.934–0.939, AVE_range_ = 0.670–0.688).

#### 2.4.3. Work Ability

We used a four-item scale to measure work ability at T1 to T3. [18]. The items are “How many points would you give your current ability to work?” and “Thinking about the [physical, mental, interpersonal] demands of your job, how do you rate your current ability to meet those demands?” Responses on these items were collected on an eleven-point scale anchored with 0 = “cannot currently work at all” and 10 = “work ability at its lifetime best.” This scale demonstrated acceptable levels of reliability at each timepoint (α_range_ = 0.904–0.932, ω_range_ = 0.904–0.934, AVE_range_ = 0.703–0.710).

#### 2.4.4. Demographics

Employee’s chronological age was measured in years since birth. Sex was coded 1 = male, 2 = female. Education was coded 1 = lower secondary school to 4 = college/university or technical college, and monthly household income was assessed in Euros (EUR) per month (see Table 1). 

### 2.5. Analysis

We initially conducted analyses to account for the (potential) influence of systematic attrition over time [34]. In summary, there was little evidence that the demographic and substantive predictors collected at T1 accounted for meaningful variance in patterns of attrition from T1 to T3 (i.e., <4% of the variance in attrition over time could be attributed to these variables; see also Table 1). Additionally, an important “first step” in modeling over-time relationships is to establish the equivalence (i.e., “invariance”) of measures collected over time [35]. In summary, we found evidence to support strong invariance over time. We additionally considered a confirmatory factor analysis on T1 measures of leadership and work ability. A five-factor model (i.e., one factor each for LMX, each of the three dimensions of age-differentiated leadership, and work ability) fit the data well (χ^2^_(314)_ = 2197, *p* < 0.001, CFI = 0.943, RMSEA = 0.072, SRMR = 0.033) and better than a one-factor model with all items specified as loading onto a single factor (Δχ^2^_(10)_ = 6519.112, *p* < 0.05). Taken together with the results of our measurement invariance models, these results bolster our confidence in the factor structure of these variables. A complete accounting of attrition and measurement equivalence analyses is available in our online appendix: https://osf.io/2bgwn/.

To account for the nesting of observations of work ability within-person, we used mixed effects modeling (implemented via the ‘lme4’ package for R) to test our focal models [36]. In terms of centering decisions, leadership variables were person–mean centered to isolate orthogonal within- and between-person components. Moreover, all between-person predictors (i.e., chronological age and between-person leadership variables) were grand mean centered prior to analysis [37]. For the ancillary models reported below, “time” was centered at T2 (i.e., parameterized as: −1, 0, 1), such that the intercept represents T2 average values of work ability.

## 3. Results

Table 2 presents intercorrelations between substantive leadership variables and work ability measured from T1 to T3. Before specifying our focal models, we considered whether there were systematic linear or non-linear changes in work ability over time by specifying an unconditional growth model. In this model, “time” was parametrized as a second-order polynomial to allow for the concurrent modeling of orthogonal time and time2 terms. This model suggests that work ability did not systematically change over time, either linearly (ɣ = −1.338, seɣ = 1.173, *p* = 0.254) or quadratically (ɣ = 0.178, seɣ = 1.214, *p* = 0.884). Despite this observation, a consideration of the intra-class correlation (ICC[1]) coefficient suggests that a notable amount of the variability observed in work ability over time occurred within-person (ICC[1] = 0.561; 1.00 − ICC[1] = 0.439 or 43.89%). Figure 1, which graphically depicts the average between- and within-person differences in T1 to T3 work ability, mirrors these findings in that although there is not a clear trend in work ability over time (i.e., suggesting neither increases nor decreases, on average) there is a notable amount of within-person variability in work ability observed.

Of note, we observed a similar pattern of ICC[1] values across the leadership variables, suggesting an appreciable amount of within-person variance over time: LMX ICC[1] = 0.716; age-differentiated leadership, balanced ICC[1] = 0.659; age-differentiated leadership, younger ICC[1] = 0.630; age-differentiated leadership, older ICC[1] = 0.636. Moreover, ICC[2] values for work ability and leadership variables ranged from ICC[2] = 0.794 to ICC[2] = 0.854, suggesting that individuals could reliably be differentiated from one another on the basis of their self-reporting of work ability and leadership variables collected over time. Given these concurring observations, we next proceeded with building our models without time as a substantive variable; however, we do report the results of further sensitivity analyses that consider the conditional effects of time below.

Table 3 presents the results of mixed effects models to test Hypotheses 1–3. With respect to Hypothesis 1, which suggested that age-differentiated leadership predicts work ability incremental to LMX, we find mixed support. Indeed, we find that between-person (ɣ = 0.335, se_ɣ_ = 0.083, *p* < 0.001) and within-person (ɣ = 0.123, se_ɣ_ = 0.058, *p* = 0.033) levels of LMX were positively related to work ability. Moreover, in support of Hypothesis 2, both between-person (ɣ = 0.358, se_ɣ_ = 0.074, *p* < 0.001) and within-person (ɣ = 0.180, se_ɣ_ = 0.046, *p* < 0.001) levels of balanced age-differentiated leadership were positively related to work ability. No other forms of age-differentiated leadership (i.e., younger- or older-focused) at either the between- or the within-person level of analysis significantly accounted for variance in work ability incremental to the contributions of LMX.

To quantify the exact contributions of each of these leadership variables to the variance explained by this model (i.e., R^2^ at both the between- and within-person levels of analysis), we conducted a dominance analysis [38]. The results suggest that 19.33% of the between-person variance (R^2^ = 0.1933) and 14.82% of the within-person variance (R^2^ = 0.1482) in work ability is explained by LMX and the three forms of age-differentiated leadership. However, at both the within- and between-person levels of analysis, between-person levels of a balanced approach to age-differentiated leadership were the dominant predictor, accounting for 6.21% (between) and 4.44% (within) of the explained variance. Between-person levels of LMX were the next most important predictor, accounting for 5.09% (between) and 3.65% (within) of the explained variance. A similar pattern (i.e., in terms of the ordering of dominant predictors) was observed when considering the contributions of within-person levels of a balanced approach to age-differentiated leadership and LMX to variance explained at the within-person level of analysis, albeit with much less (i.e., <1% total) of the variance accounted for. Figure 2 depicts a graphical representation of this variance-explained decomposition, at both levels of analysis and for between- and within-person predictors.

With respect to Hypothesis 3, which suggested that older (younger) workers benefit more than younger (older) workers from age-differentiated leadership focused on older (younger) workers, we next considered a model with work ability regressed onto age-by-leadership interactions (i.e., three interactions representing age-by-age-differentiated leadership; age-by-LMX). Partially supporting our hypothesis, a significant interaction between age and between-person levels of age-differentiated leadership focused on older workers was observed (ɣ = 0.019, se_ɣ_ = 0.009, *p* = 0.032). To understand the nature of this interaction, we computed Johnson–Neyman regions of significance, which can be interpreted around the mean age of the sample (M = 44.18 years old) and its standard deviation (SD = 10.84 years old). The results suggest that the slope of the relationship defining between-person levels of age-differentiated leadership focused on older workers and work ability is significant and positive for older workers (i.e., those aged 16.60 years above the mean age of the sample and higher; see also Figure 3A).

Moreover, we also observed two significant interaction effects that were not hypothesized. First, we found a significant interaction between age and between-person levels of a balanced approach to age-differentiated leadership (ɣ = −0.019, se_ɣ_ = 0.007, *p* = 0.005). Johnson–Neyman regions of significance suggest that the slope of the relationship defining between-person levels of a balanced approach to age-differentiated leadership and work ability is significant and positive for average age (i.e., <+1 S.D. above the mean) and younger workers (i.e., those aged 9.39 years above the mean age of the sample and lower; see Figure 3B).

Second, we also found a significant interaction between age and between-person levels of LMX (ɣ = −0.022, se_ɣ_ = 0.008, *p* = 0.004). Johnson–Neyman regions of significance suggest that the slope of the relationship defining between-person levels of LMX and work ability is significant and positive for average age (i.e., <+1 S.D. above the mean) and younger workers (i.e., those aged 7.41 years above the mean age of the sample and lower; see Figure 3C).

### Sensitivity Analysis

As suggested, our initial analysis showed that there were no systematic effects of time (linear, quadratic) on work ability. However, given the nesting of observations within-person over time, we additionally ran mixed effects models that (a) controlled for time, and (b) considered time-graded slopes of leadership on work ability. In the former case, controlling for time did not change the substantive conclusions drawn here with respect to main or interaction effects. Thus, as suggested above, and in order to obtain cleaner estimates of explained variance (i.e., irrespective of the influence of time as a covariate), we omitted time from the models presented in Table 3. Moreover, no significant time-by-leadership or time-by-age-by-leadership interactions were observed. Again, these interactions were omitted here largely for the sake of parsimony; however, complete results of these analyses are available in our online appendix: https://osf.io/2bgwn/.

## 4. Discussion

Based on the idea that leadership may benefit the healthy aging of employees of different ages, we conducted the first study that examined the influence of age-differentiated leadership and LMX on work ability. We found some, albeit mixed support for our first and second hypotheses: that age-differentiated leadership incrementally accounts for variance in work ability above-and-beyond LMX and that a balanced approach to age differentiated leadership benefits workers of all ages. Indeed, only a balanced approach to age-differentiated leadership was found to significantly account for variance in work ability, and still the relative contribution of this form of age-differentiated leadership was only slightly more than that of LMX at both the within- and between-person levels of analysis.

Regarding our third hypothesis, we found partial support for the prediction of age-conditional effects of age-differentiated leadership on work ability. Specifically, we found that relatively older workers benefit more than relatively younger workers from age-differentiated leadership that focuses on older employees. Thus, leadership that considers older workers’ changing needs, such as a greater interest in autonomy and the use of acquired skills [23], may promote healthy aging at work. Moreover, although not hypothesized, we found that employees just over the average age of the sample and younger were more likely to benefit from a balanced approach to age-differentiated leadership. No significant age-graded effects of age-differentiated leadership that focuses on younger employees were observed. This observation might be due to the on-average “older” age of our sample (i.e., M = 44.18 years old; SD = 10.84 years old). Indeed, it could be the case that in samples that include a larger proportion of relatively younger employees, there would be additional benefits (i.e., in terms of work ability, but also other health, wellbeing, and motivation-related outcomes) of age differentiated leadership focused on younger employees.

More broadly, the findings of this study suggest that, over time, an approach to leadership that balances the needs of workers of all ages is optimal for bolstering the work ability of most employees (i.e., including those just over the average age of the sample and younger), whereas an approach to leadership that focuses on the needs of older workers is optimal for bolstering the work ability of relatively older employees. Given the context of the study (i.e., with data collected before and during the first wave of the COVID-19 pandemic), the observation that a balanced approach to age differentiated leadership was effective at bolstering work ability for a large subset of the sample might suggest that leaders attempted to balance the needs of workers of all ages while navigating the uncertainty associated with the early stages of the pandemic.

Additionally, and unexpectedly, we also find that age and LMX interacted to predict work ability. The work ability of employees just over the average age of the sample and younger seems to benefit more from LMX than relatively older employees. This finding could be explained with socioemotional selectivity theory [38], which suggests that due to their expansive future time perspective, relatively younger employees prioritize instrumental goals at work. These needs may be satisfied by establishing and maintaining a positive relationship with one’s direct supervisor (i.e., fostering a more positive leader–member exchange). Although interesting, more theory-driven research would be needed to unpack the specific goal-relevant mechanism that helps to explain this finding (see, e.g., the lifespan perspective on leadership and followership [9]).

The finding that, on average, work ability was stable between December 2019 and June 2020 is interesting. Common wisdom surrounding the COVID-19 pandemic is that it has been a particularly challenging time and that people have experienced increased demands, which would otherwise be considered strains on work ability. These findings run contrary to the common narrative that the early stages of the pandemic (and the lockdown periods that ensued, e.g., the first national lockdown in Germany occurred between mid-March and early May 2020) severely curtailed people’s capacity to manage the demands of their jobs. While this was certainly the case for a proportion of employees in our sample (i.e., within-person decreases in work ability), others reported increases in or maintenance of work ability. It may be the case that the latter worked in jobs that were not strongly affected by the pandemic and lockdown [32]. Moreover, our findings that LMX and different forms of age-differentiated leadership show positive unconditional and age-conditional relationships with work ability suggest that these forms of leadership can be particularly beneficial for buoying employee wellbeing, and supporting healthy aging at work, during times of crisis.

### 4.1. Theoretical and Practical Implications

Our findings generally support the idea that age-differentiated work systems contribute to successful and healthy aging at work [13,16]. Specifically, our results partially support emerging thought on the role of age-differentiated leadership [12,25]. Indeed, as cautioned more generally in the leadership literature, the development of new leadership constructs needs to build on existing and well-established ones [39]. Our results suggest that a balanced approach to age-differentiated leadership is beneficial to work ability incremental to LMX. Moreover, although there has been a great deal of research on the relationship between leadership and employee wellbeing, research has thus far largely neglected work ability as an outcome of leadership, and research has not considered how employees of different ages may benefit from different forms of leadership with respect to their work ability. Indeed, even though Ilmarinen’s [40] conceptual model of work ability considers leadership as a key predictor, recent empirical research on work ability has focused on health, a sense of control, job demands, and job and personal resources as antecedents of work ability and neglected leadership behaviors [18].

In terms of practical implications, these results suggest that encouraging a balanced approach to age-differentiated leadership that considers the needs of employees of all ages while recognizing the specific needs of older employees, coupled with positive LMX relationships, may be a benefit to the wellbeing of workers of all ages. To this end, leadership development programs should consider including elements of LMX and age-differentiated leadership in attempts to promote leadership behavior that beneficially serves the needs of employees of all ages and across the lifespan. Indeed, this strategy is conceptually supported in that recent research suggests that, similar to other forms of leadership [41], leadership that considers the needs of employees of all ages can be trained successfully [11,42].

### 4.2. Limitations and Directions for Future Research

This study has several strengths, especially that we collected data within-person and over a time frame of six months. Nevertheless, it also has a few limitations. First, leadership variables and work ability were collected exclusively via self-reports, which may raise concerns about rater biases and inflated associations between leadership and work ability assessments. Future research could obtain behavioral measures of leadership [43] and/or coworker or supervisor ratings of work ability [44].

Second, as suggested, although there was an appreciable amount of within-person variability in work ability to be modeled, there were no observed effects of time in our analysis. Thus, while our results speak to the importance of differentiating between-person from within-person relationships, this study cannot speak to how changes in leadership perceptions over time affect work ability, or how the correspondence between leader and employee perceptions of leadership influence such processes. Likewise, although our study was conducted during the pandemic, we have not explicitly modeled any pandemic-specific experiences that might help to explain the variability we observe in work ability across this time frame. However, the observation of an appreciable amount of within-person variability in work ability over time (i.e., 43.89%) begs for additional theoretical and empirical elaboration. Future research to this end should consider the timeframe across which work ability is likely to vary systematically with time (i.e., in terms of *trajectories* of work ability that vary as a function of time) by adopting a continuous time approach [45]. Additionally, as research has found that work ability can fluctuate within-person and over time even in the short term (e.g., within-day [46]), research should adopt daily diary designs to better understand the consequences of such variability.

Third, and related to the second point regarding timing, the timeframe of data collected (i.e., December 2019 to June 2020) coincided with the onset and early stages of the COVID-19 pandemic in Germany. As offered, our results suggest relative stability, on average, in work ability during this timeframe. Future research should consider multiple reports of leader and employee perceptions of LMX and age-differentiated leadership over time to allow such dynamic and reciprocal relationships to be studied as emergent phenomena [47]. To this end, too, research should further consider how leadership and wellbeing co-occurred across the pandemic. Research should also consider the influence of age differentiated leadership and LMX in comparison to other established leadership constructs, which might prove to differentially benefit workers of different ages and lend further support to the role of leadership in the promotion of healthy aging at work. To this end, the focus should especially be on leadership constructs that, similar to age differentiated leadership and LMX, focus on individual needs and promoting positive relationships between leaders and employees [9].

Finally, we focused on only one outcome, work ability, in this study. Although work ability is an important index of healthy aging at work, and salient within the context of the pandemic, future research should also consider a broader array of outcomes that may be more sensitive to age-differentiated leadership (e.g., follower perceptions of leader effectiveness, follower satisfaction with leader, follower extra effort [48]; perceptions of age-diversity climate [49]).

## 5. Conclusions

Using a three-wave longitudinal research design, we examined the role that age-differentiated leadership and LMX played in the prediction of work ability during the early stages of the COVID-19 pandemic [50]. The results suggest that work ability did not change systematically over time during the early stages of the pandemic, but did exhibit a notable degree of within-person variability over time. Additionally, we find that a balanced approach to age-differentiated leadership (i.e., one that considers the needs of both older and younger employees) matters for predicting within-person variability in work ability, and complements the positive influence of LMX toward this end. Moreover, older employees’ work ability particularly benefited from an approach to age-differentiated leadership that considers older employees’ needs, whereas younger employees’ work ability especially benefited from LMX and a balanced approach to age-differentiated leadership. Overall, these results provide initial support for the idea that an age-differentiated approach to leadership contributes to healthy aging at work.

## Figures and Tables

**Figure 1 ijerph-18-12509-f001:**
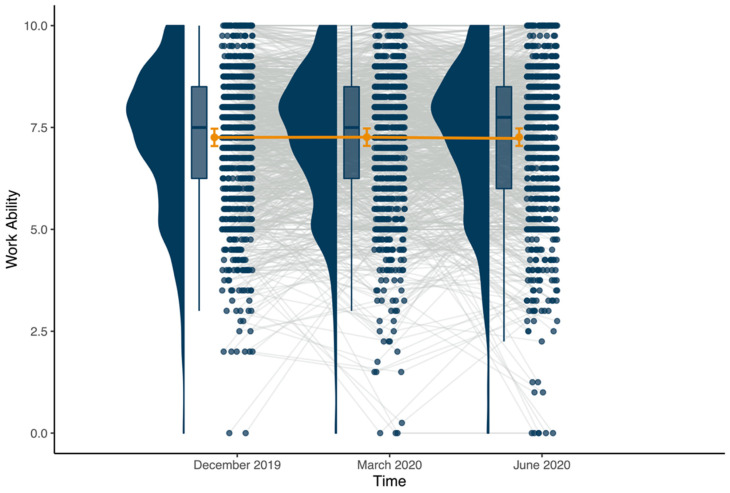
Between- and Within-Person Variability in Work Ability from T1 (December 2019) to T3 (June 2020). Note: Connected orange points and associated 95% confidence intervals represent between-person relationships; connected blue points represent within-person relationships.

**Figure 2 ijerph-18-12509-f002:**
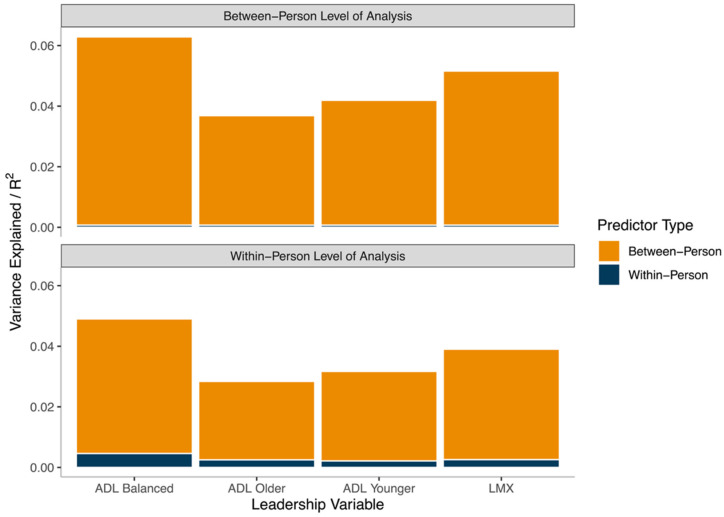
Results of Dominance Analysis. Note: ADL = age-differentiated leadership; LMX = leader-member exchange. Total within-person R^2^ = 0.1482; total between-person R^2^ = 0.1933. Estimates of R^2^ derived using formulae from Snijders and Bosker.

**Figure 3 ijerph-18-12509-f003:**
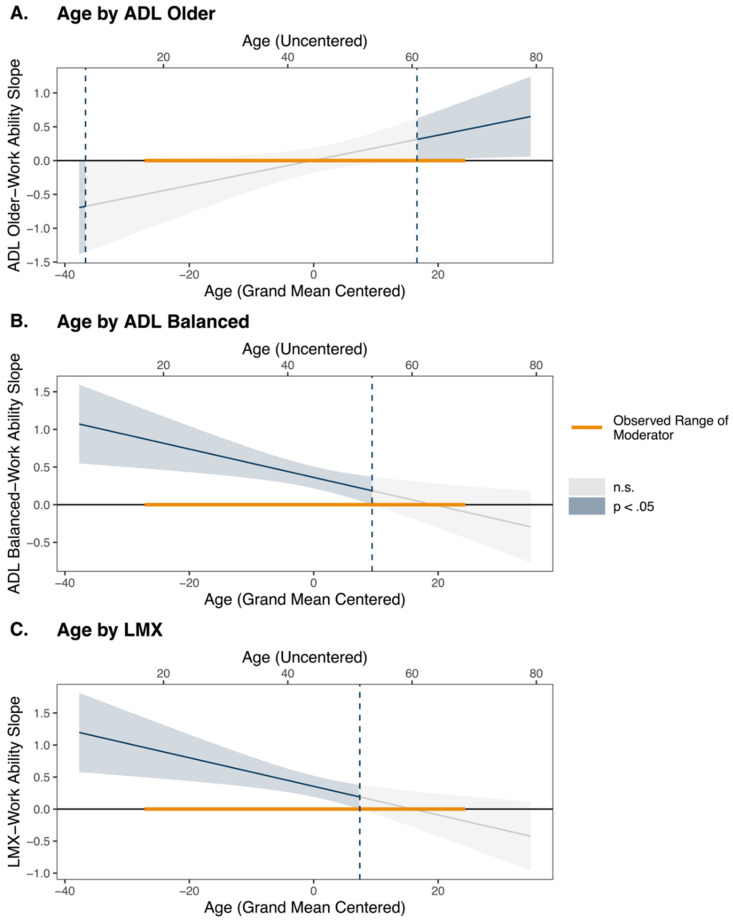
Johnson–Neyman Significance Regions Representing Age-Conditional Relationships of Leadership on Work Ability. Note. ADL = age-differentiated leadership; LMX = leader-member exchange. Chronological age and between-person leadership predictors (i.e., ADL Older, ADL balanced, and LMX) are grand mean centered.

**Table 1 ijerph-18-12509-t001:** Demographics and Descriptive Statistics.

	Incomplete Sample	Complete Sample	
	(*n* = 1831)	(*n* = 1145)	*p*-Value
**Sex**			
Male	327 (17.9%)	652 (56.9%)	<0.001
Female	435 (23.8%)	488 (42.6%)	
Missing	1069 (58.4%)	5 (0.4%)	
**Age (Years)**			
Mean (SD)	43.0 (12.7)	44.9 (10.8)	0.0012
Median (Min, Max)	43.0 (19.0, 99.0)	46.0 (18.0, 69.0)	
Missing	1064 (58.1%)	1 (0.1%)	
**Education**			
Lower Secondary School	67 (3.7%)	77 (6.7%)	0.129
Intermediate Secondary School	267 (14.6%)	420 (36.7%)	
Upper Secondary School	146 (8.0%)	187 (16.3%)	
College/University or Technical College	280 (15.3%)	448 (39.1%)	
Missing	1071 (58.5%)	13 (1.1%)	
**Monthly Household Income (€)**			
0–999	90 (4.9%)	58 (5.1%)	<0.001
1000–1999	139 (7.6%)	182 (15.9%)	
2000–2999	161 (8.8%)	267 (23.3%)	
3000–3999	152 (8.3%)	250 (21.8%)	
4000–4999	121 (6.6%)	202 (17.6%)	
5000–5999	58 (3.2%)	96 (8.4%)	
6000–6999	46 (2.5%)	89 (7.8%)	
Missing	1064 (58.1%)	1 (0.1%)	
**T1 Leader-Member Exchange**			
Mean (SD)	3.33 (0.955)	3.33 (0.933)	0.998
Median (Min, Max)	3.36 (1.00, 5.00)	3.43 (1.00, 5.00)	
Missing	1449 (79.1%)	3 (0.3%)	
**T1 ADL Balanced**			
Mean (SD)	4.80 (1.49)	4.83 (1.49)	0.721
Median (Min, Max)	5.00 (1.00, 7.00)	5.00 (1.00, 7.00)	
Missing	1449 (79.1%)	3 (0.3%)	
**T1 ADL Older**			
Mean (SD)	4.51 (1.48)	4.51 (1.50)	0.961
Median (Min, Max)	4.40 (1.00, 7.00)	4.40 (1.00, 7.00)	
Missing	1449 (79.1%)	3 (0.3%)	
**T1 ADL Younger**			
Mean (SD)	4.50 (1.50)	4.51 (1.50)	0.887
Median (Min, Max)	4.50 (1.00, 7.00)	4.50 (1.00, 7.00)	
Missing	1449 (79.1%)	3 (0.3%)	
**T1 Work Ability**			
Mean (SD)	7.32 (1.68)	7.27 (1.70)	0.611
Median (Min, Max)	7.75 (−1.00, 10.0)	7.50 (0, 10.0)	
Missing	1390 (75.9%)	1 (0.1%)	

**Table 2 ijerph-18-12509-t002:** Descriptive Statistics and Intercorrelations Among Substantive Variables.

	Variable	M	SD	1.	2.	3.	4.	5.	6.	7.	8.	9.	10.	11.	12.	13.	14.	15.
1.	T1 ADL Bal.	4.84	1.47	(0.96)														
2.	T1 ADL Older	4.53	1.48	0.85	(0.94)													
3.	T1 ADL Young.	4.53	1.49	0.83	0.87	(0.94)												
4.	T2 ADL Bal.	4.92	1.51	0.65	0.61	0.59	(0.96)											
5.	T2 ADL Older	4.60	1.52	0.60	0.65	0.61	0.88	(0.95)										
6.	T2 ADL Young.	4.59	1.51	0.57	0.61	0.63	0.83	0.88	(0.95)									
7.	T3 ADL Bal.	4.85	1.48	0.64	0.59	0.59	0.68	0.62	0.61	(0.96)								
8.	T3 ADL Older	4.60	1.49	0.60	0.61	0.59	0.64	0.65	0.63	0.89	(0.95)							
9.	T3 ADL Young.	4.56	1.49	0.57	0.58	0.61	0.62	0.63	0.66	0.84	0.89	(0.95)						
10.	T1 LMX	3.32	0.93	0.72	0.73	0.73	0.59	0.60	0.58	0.58	0.58	0.58	(0.93)					
11.	T2 LMX	3.39	0.95	0.57	0.57	0.58	0.72	0.74	0.73	0.60	0.61	0.61	0.70	(0.93)				
12.	T3 LMX	3.38	0.94	0.58	0.59	0.60	0.60	0.62	0.61	0.76	0.76	0.74	0.71	0.74	(0.94)			
13.	T1 Wrk. Ability	7.26	1.70	0.35	0.32	0.32	0.30	0.28	0.26	0.34	0.31	0.31	0.34	0.31	0.32	(0.91)		
14.	T2 Wrk. Ability	7.28	1.70	0.30	0.29	0.28	0.37	0.33	0.33	0.36	0.33	0.32	0.31	0.34	0.32	0.61	(0.90)	
15.	T3 Wrk. Ability	7.25	1.80	0.27	0.26	0.25	0.32	0.29	0.26	0.39	0.34	0.33	0.30	0.32	0.35	0.53	0.60	(0.93)

Note. ADL = age-differentiated leadership; LMX = leader-member exchange; Bal. = balanced; Young. = younger; Wrk. = work. Coefficient alpha reliabilities are presented along the diagonal. Descriptives and correlations based on *n =* 968 complete respondents (listwise deletion); see online appendix (https://osf.io/2bgwn/) for more details. For all correlations, *p* < 0.05.

**Table 3 ijerph-18-12509-t003:** Summary of Mixed Effects Models.

	Work Ability			Work Ability		
*Predictors*	*ɣ*	*se_ɣ_*	*p*	*ɣ*	*se_ɣ_*	*p*
(Intercept)	7.246	0.039	<0.001	7.259	0.039	<0.001
LMX Btwn.	0.335	0.083	<0.001	0.355	0.084	<0.001
ADL Balanced Btwn.	0.358	0.074	<0.001	0.362	0.074	<0.001
ADL Younger Btwn.	−0.050	0.081	0.535	−0.082	0.082	0.322
ADL Older Btwn.	−0.015	0.094	0.871	0.004	0.095	0.964
LMX Wthn.	0.123	0.058	0.033	0.125	0.058	0.031
ADL Balanced Wthn.	0.180	0.046	<0.001	0.182	0.046	<0.001
ADL Younger Wthn.	0.050	0.043	0.245	0.043	0.043	0.321
ADL Older Wthn.	−0.058	0.048	0.226	−0.052	0.048	0.282
Age				0.007	0.004	0.050
Age × LMX Btwn.				−0.022	0.008	0.004
Age × ADL Balanced Btwn.				−0.019	0.007	0.005
Age × ADL Younger Btwn.				0.004	0.008	0.581
Age × ADL Older Btwn.				0.019	0.009	0.032
Age × LMX Wthn.				0.000	0.005	0.950
Age × ADL Balanced Wthn.				0.001	0.004	0.796
Age × ADL Younger Wthn.				0.001	0.004	0.727
Age × ADL Older Wthn.				−0.008	0.004	0.074
SD (Intercept)	1.138			1.121		
SD (Observations)	1.065			1.065		
**Random Effects**						
σ^2^	1.290			1.280		
τ_00_	1.290			1.260		
ICC	0.500			0.490		
N	1133			1132		
Observations	3236			3235		
Within R^2^/Between R^2^	0.148/0.193			0.161/0.211		

Note. ADL = age-differentiated leadership; LMX = leader-member exchange. Btwn. = between; Wthn. = within.

## Data Availability

De-identified data on which the study conclusions are based are available in our online appendix https://osf.io/2bgwn/.

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
