# Peer review of "Age-Differentiated Leadership and Healthy Aging at Work: Evidence from the Early Stages of the COVID-19 Pandemic"

_ijerph, 2021, doi:10.3390/ijerph182312509_

Round 1
Reviewer 1 Report
An interesting article of this prolific group of authors. Having a look to their production and self-citations, this article fits adequately with their global ideas and main conclusions about the effects of the first stages of the pandemic.
Minor commentaries from this reviewer:
1. It is not necessary to insert “see” before references e.g.(see [10,11]). Revise all text carefully, as some citations are correct, e.g (e.g., see Kooij et al.’s [13] perspective on “successful aging at work”).
2. The Introduction has a strange cohesion:
- The aims and objectives are presented from lines 60-76. Previous information is well presented.
- Information between lines 77-131 (2 subsections) is quite long. Why is this information not integrated in the introduction and before the objectives (lines 28-76)? I suggest omitting these subsections.
- The hypotheses are located after the objectives. Should they have a subsection?
Author Response
Comments from Reviewer #1
- An interesting article of this prolific group of authors. Having a look to their production and self-citations, this article fits adequately with their global ideas and main conclusions about the effects of the first stages of the pandemic.
Author Response: We are very pleased to hear that you think our ideas and conclusions inform a better understanding of the early stages of the COVID-19 pandemic. Thank you for your comments on our manuscript, which we have addressed, below.
- It is not necessary to insert “see” before references e.g.(see [10,11]). Revise all text carefully, as some citations are correct, e.g (e.g., see Kooij et al.’s [13] perspective on “successful aging at work”).
Author Response: We have carefully edited the manuscript for such style concerns, with a particular eye on citation style.
- The Introduction has a strange cohesion:
- The aims and objectives are presented from lines 60-76. Previous information is well presented.
- Information between lines 77-131 (2 subsections) is quite long. Why is this information not integrated in the introduction and before the objectives (lines 28-76)? I suggest omitting these subsections.
- The hypotheses are located after the objectives. Should they have a subsection?
Author Response: We have reorganized these subsections as suggested. Specifically, per your advice, we have introduced earlier the sections “Healthy Aging at Work” and “Age-Differentiated Leadership and Employee Work Ability” (now Section 1.2. and 1.3., respectively). Section 1.3. now provides an overview of study objectives and precedes section 1.4. where we present our hypotheses.
Reviewer 2 Report
This paper examines the relative influence of age-differentiated leadership on healthy aging at work based on a three-wave longitudinal study conducted during the early stages of COVID-19 pandemic. The authors mainly find that work ability did not change systematically over time during the early stages of the pandemic but did show notable within-person variability. They show that a balanced approach to age-differentiated leadership matters for predicting within-person variability in work ability. Older employees’ work ability especially benefited from an approach to age-differentiated leadership as well. This paper is generally well-structured and has significant contribution by the impact of age-differentiated leadership. I have a few comments on this paper.
- Within Work Ability Variability: This paper does not seem to clearly explain the reason behind its one of major findings. What factors potentially determine within ability variability without systemic ones? Did the pandemic causes the variability?
- The title is misleading: The title of this paper suggests that this paper tests how COVID-19 may change the extant relationship between ADL and other work related variables. However, the hypotheses in this paper is a general one and is not dependent on the condition of COVID-19. It may be better to develop a COVID-19 dependent hypothesis or to deemphasize the meaning of COVID-19.
- Minor Correction: The paper may not exactly follow the MDPI style in the introduction section. I believe that the introduction section does not use the name of authors to refer previous studies. But this paper use Kooij et al.’s [13] in line 52. Please check the reference style.
Author Response
Comments from Reviewer #2
This paper examines the relative influence of age-differentiated leadership on healthy aging at work based on a three-wave longitudinal study conducted during the early stages of COVID-19 pandemic. The authors mainly find that work ability did not change systematically over time during the early stages of the pandemic but did show notable within-person variability. They show that a balanced approach to age-differentiated leadership matters for predicting within-person variability in work ability. Older employees’ work ability especially benefited from an approach to age-differentiated leadership as well. This paper is generally well-structured and has significant contribution by the impact of age-differentiated leadership. I have a few comments on this paper.
Author Response: Thank you for your apt summary of our work. We are glad to hear that you found our paper to be well structured and to make a significant contribution.
- Within Work Ability Variability: This paper does not seem to clearly explain the reason behind its one of major findings. What factors potentially determine within ability variability without systemic ones? Did the pandemic causes the variability?
Author Response: Thank you for raising this important point. Indeed, various features of the pandemic (e.g., experiencing lockdowns) might have contributed to within-person variability in work ability. Importantly, too, research has shown that work ability can fluctuate within-person and in the short term. We now point this out in text (see pages 13-14, lines 442-453). Please note that our statistical models for tests of Hypotheses 1-3 do pertain to a set of theoretically relevant variables that systematically account for variability in work ability.
- The title is misleading: The title of this paper suggests that this paper tests how COVID-19 may change the extant relationship between ADL and other work related variables. However, the hypotheses in this paper is a general one and is not dependent on the condition of COVID-19. It may be better to develop a COVID-19 dependent hypothesis or to deemphasize the meaning of COVID-19.
Author Response: We appreciate this comment regarding the title. To better reflect the intended message that our study took place during the COVID-19 pandemic without being misleading regarding what we tested, we have changed the title to “Age-Differentiated Leadership and Healthy Aging at Work: Evidence from the Early Stages of the COVID-19 Pandemic.”
- Minor Correction: The paper may not exactly follow the MDPI style in the introduction section. I believe that the introduction section does not use the name of authors to refer previous studies. But this paper use Kooij et al.’s [13] in line 52. Please check the reference style.
Author Response: We have carefully edited the manuscript for such style concerns, with a particular eye on citation style.
Reviewer 3 Report
The article deals with an interesting and actual topic. It gives new perspective of workforce ageing implications.
The aims defined clearly. The research made it possible to achieve them.
The research methods have been selected, described and used properly. Interesting presentation of the results and their discussion.
Authors are aware of the research limitations, nevertheless I would suggest to develop that part of the article.
Is it possible to formulate so general conclusions basing on those research results? Specificity of COVID19 situation and different solutions implemented in different countries/ companies could reflect/subjectivize the results. Authors have focused on very narrow approach to the issue of healthy ageing at work. Maybe the article's title is too extensive (the reader expects a broader approach to the issue than it really is), and should be limited to "work ability" - for authors decision.
Author Response
Comments from Reviewer #3
- The article deals with an interesting and actual topic. It gives new perspective of workforce ageing implications.
- The aims defined clearly. The research made it possible to achieve them.
- The research methods have been selected, described and used properly. Interesting presentation of the results and their discussion.
Author Response: We are very pleased to hear that you think our paper is interesting and pertains to a current topic with important implications for the aging workforce. We are also very happy to hear that the aims of our study are clear and that the methods are appropriate to support our results.
- Authors are aware of the research limitations, nevertheless I would suggest to develop that part of the article.
Author Response: We have now been careful to edit this section to further develop it in light of this comment and comments by other reviewers (see pages 13-14, lines 442-453).
- Is it possible to formulate so general conclusions basing on those research results? Specificity of COVID19 situation and different solutions implemented in different countries/ companies could reflect/subjectivize the results. Authors have focused on very narrow approach to the issue of healthy ageing at work. Maybe the article's title is too extensive (the reader expects a broader approach to the issue than it really is), and should be limited to "work ability" - for authors decision.
Author Response: We appreciate this comment regarding the title. Our study does consider processes of age-differentiated leadership and healthy aging in situ during the early stages of the COVID-19 pandemic. However, in line with your comments, to better reflect the intended message that our study took place during the COVID-19 pandemic without being misleading regarding what we modeled or the scope of our conclusions thereto, we have changed the title to “Age-Differentiated Leadership and Healthy Aging at Work: Evidence from the Early Stages of the COVID-19 pandemic.” Given the focus of this special issue as such, we think this is an appropriate framing for our study and we are careful to caveat our interpretation of our findings accordingly.
Moreover, as our core theorizing pertains to healthy aging at work (of which work ability is but one possible indicator), we have chosen to keep this framing in the title as well.
Round 2
Reviewer 2 Report
This paper has improved significantly by reflecting the comments and seems qualified for publication.